# The Ndc80 complex bridges two Dam1 complex rings

Jae ook Kim[1], Alex Zelter[1], Neil T Umbreit[1†], Athena Bollozos[1], Michael Riffle[1], Richard Johnson[2], Michael J MacCoss[2], Charles L Asbury[3], Trisha N Davis[1*]

[1]Department of Biochemistry, University of Washington, Seattle, United States; [2]Department of Genome Sciences, University of Washington, Seattle, United States; [3]Department of Physiology and Biophysics, University of Washington, Seattle, United States

**Abstract** Strong kinetochore-microtubule attachments are essential for faithful segregation of sister chromatids during mitosis. The Dam1 and Ndc80 complexes are the main microtubule binding components of the *Saccharomyces cerevisiae* kinetochore. Cooperation between these two complexes enhances kinetochore-microtubule coupling and is regulated by Aurora B kinase. We show that the Ndc80 complex can simultaneously bind and bridge across two Dam1 complex rings through a tripartite interaction, each component of which is regulated by Aurora B kinase. Mutations in any one of the Ndc80p interaction regions abrogates the Ndc80 complex's ability to bind two Dam1 rings in vitro, and results in kinetochore biorientation and microtubule attachment defects in vivo. We also show that an extra-long Ndc80 complex, engineered to space the two Dam1 rings further apart, does not support growth. Taken together, our work suggests that each kinetochore in vivo contains two Dam1 rings and that proper spacing between the rings is vital.

*For correspondence: tdavis@uw.edu

Present address: †Department of Pediatric Oncology, Dana-Farber Cancer Institute, Boston, United States

Competing interests: The authors declare that no competing interests exist.

## Introduction

Kinetochores link replicated chromosomes to spindle microtubules. They form attachments flexible and strong enough to stay attached to microtubules during assembly and disassembly. Estimates of the strength of the attachments suggest that they are much stronger than required just to pull a chromosome through the cellular milieu (*Fisher et al., 2009*; *Nicklas, 1965*). Yet, incorrect attachments are weak enough to be detached and corrected.

The Dam1 and Ndc80 complexes are the main microtubule-binding components in the budding yeast kinetochore. The Ndc80 complex connects the Dam1 complex to the rest of the kinetochore. The two complexes interact on microtubules, significantly increasing the strength of Ndc80 complex attachment to microtubules (*Lampert et al., 2010, 2013*; *Tien et al., 2010*). How these two complexes interact on microtubules remains uncertain with conflicting results reported in the literature (*Lampert et al., 2013*; *Maure et al., 2011*).

In addition to providing strong attachments between chromosomes and spindle microtubules, kinetochores also serve as regulatory hubs. The Ndc80 complex and the Dam1 complex are at the center of this regulation. The Ndc80 complex is a scaffold for the spindle assembly checkpoint (*Dou et al., 2015*; *Hiruma et al., 2015*; *Ji et al., 2015*). Both the Dam1 and Ndc80 complexes are targets of Aurora B kinase, which corrects aberrant kinetochore-microtubule attachments to achieve bioriented attachments (Reviewed in *Sarangapani and Asbury, 2014*). Aurora B kinase phosphorylation of the Dam1 complex components Dam1p, Ask1p, and Spc34p together disrupts the interaction between the Dam1 and Ndc80 complexes (*Lampert et al., 2013*; *Tien et al., 2010*); however, which phospho-protein is responsible for disrupting the interaction has yet to be deciphered.

**eLife digest** The genetic material inside yeast, human and other eukaryotic cells is stored within structures called chromosomes. Every time a cell divides to make two "daughter" cells, all the chromosomes in the cell must be copied and then separated into two equal sets, with each set delivered to a different daughter cell. If the copied chromosomes divide into unequal sets the daughter cells may die or not work properly.

Chromosomes are attached to tube-like structures called microtubules, which pull the two sets of chromosomes to opposite ends of the cell just before it divides. Microtubules are constantly shrinking and growing, and chromosomes must stay attached the whole time, or they will not be correctly separated. Many proteins are involved in attaching chromosomes to microtubules, including two groups known as the Dam1 complex and the Ndc80 complex. The Dam1 complex forms a ring around microtubules, while the Ndc80 complex forms a long flexible rod that can bend in the middle. However, it was not clear how these shapes allow these complexes to perform their roles in cells.

Kim et al. studied these two complexes from yeast cells. The experiments show that the Ndc80 rod bridges two Dam1 rings, not one as previously assumed. Both of these rings and the rod are required for chromosomes to attach to microtubules. Cells with defective Ndc80 rods that can only bind to one ring do not distribute their chromosomes correctly when they divide. Kim et al. also show that the Ndc80 rod holds the Dam1 rings a specific distance apart, which also appears to be important for the chromosomes to be correctly divided between daughter cells.

The next step following on from this work is to find out exactly why both rings are needed and why they seem to need to be a set distance apart. Human and other eukaryotic cells divide in a similar way to yeast cells, so these findings may help us to understand what goes wrong in Down's syndrome and other diseases caused by cells having the wrong number of chromosomes.

The spontaneous assembly in vitro of the Dam1 complex into microtubule-encircling rings suggests a compelling mechanism for kinetochore-microtubule coupling. The ring might be pushed by the curled depolymerizing microtubule protofilaments and prevented from falling off of flared assembling ends (*Asbury et al., 2011*; *McIntosh et al., 2008*). We don't know the organization of the Dam1 complex in the cell, although oligomerization is essential for formation of strong kinetochore-microtubule attachments (*McIntosh et al., 2013*; *Umbreit et al., 2014*). Estimates of the number of Dam1 complexes at the kinetochore in vivo have suggested numbers as low as 10, which is not enough to form a single 16-membered ring, and as high as 32, which is enough for two rings (*Aravamudhan et al., 2013*; *Joglekar et al., 2006*; *Lawrimore et al., 2011*). Despite this uncertainty, previous budding yeast kinetochore models have assumed one Dam1 complex ring (*Aravamudhan et al., 2015*; *Cheeseman and Desai, 2008*; *Joglekar et al., 2009*; *Tanaka et al., 2007*).

In this study, we provide evidence that the kinetochore requires the Ndc80 complex to bind and bridge two Dam1 complex rings in vivo and that Aurora B kinase regulates the interactions at both rings. Our work suggests that the specific ring-ring distance, defined by the Ndc80 complex bridge, is important for supporting growth. The presence of two Dam1 rings at the kinetochore, in a specific orientation bound to the Ndc80 complex, would have implications for how attachment strength is established and modulated by tension.

## Results

### Three Ndc80p regions interact with Dam1p, Ask1p, and Spc34p of the Dam1 complex

We identified where the Dam1 and Ndc80 complexes interact using protein cross-linking and mass spectrometry analysis (*Figure 1A* and *Figure 1—figure supplement 1*) following protocols developed previously (*Hoopmann et al., 2015*; *Kudalkar et al., 2015*; *Tien et al., 2014*; *Zelter et al., 2015*). The full decameric Dam1 complex and tetrameric Ndc80 complex were cross-linked in the

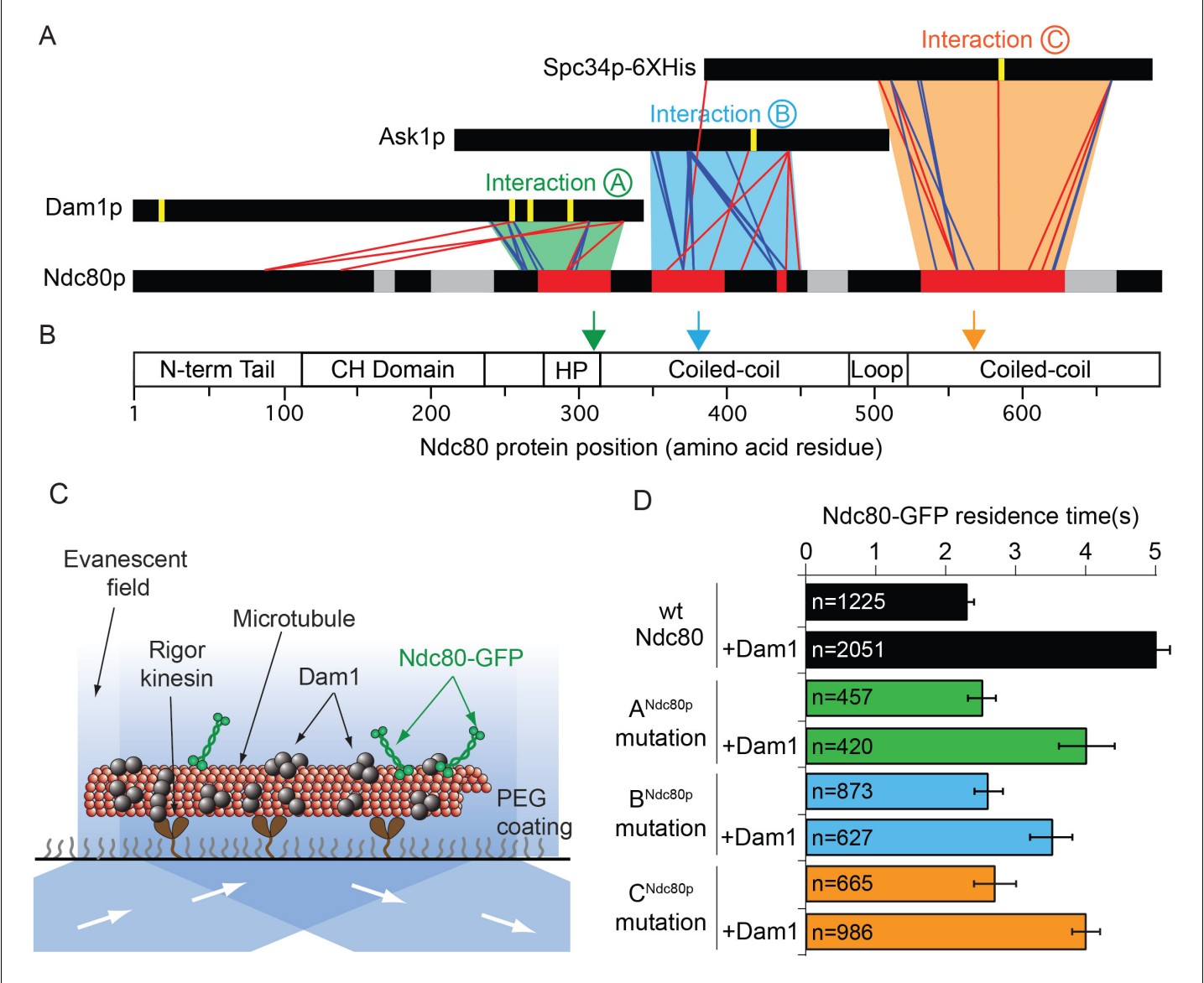

**Figure 1.** Dam1p, Ask1p, and Spc34p form cross-links to three distinct Ndc80p regions. (**A**) Cross-links between the Dam1p, Ask1p, and Spc34p of the Dam1 complex and Ndc80p of the Ndc80 complex. Dam1 and Ndc80 complexes were cross-linked in the presence of microtubules. Horizontal black bars represent proteins and the six vertical yellow lines indicate Aurora B kinase phosphorylation sites on the Dam1 complex. Red and blue lines show cross-links formed with DSS and EDC cross-linkers, respectively. For clarity, only the cross-links between the Dam1 complex proteins and Ndc80p are displayed. Data are shown for peptides with Percolator (**Käll et al., 2007**) assigned q-values $\leq$ 0.05. Red bars on Ndc80p indicate regions where clusters of lethal mutations mapped (from **Tien et al., 2013**) to cross-linked regions. Grey bars on Ndc80p indicate clusters of lethal insertions outside of cross-linked regions. Green, blue, and orange trapezoids represent putative interactions (**A**, **B**, and **C**) between the Dam1 and Ndc80 complexes. (**B**) Bar diagram of Ndc80p with structural features. Green, blue and orange arrows indicate the positions of lethal mutations in interaction regions A[Ndc80p], B[Ndc80p], and C[Ndc80p] used in this study. (CH: calponin homology; HP: hairpin) (**C**) Diagram showing the setup of TIRF microscopy experiments. Single molecule Ndc80-GFP complex binding on microtubules was visualized in the presence or absence of the Dam1 complex. (**D**) Lethal mutation in region A[Ndc80], B[Ndc80], or C[Ndc80] partially disrupts the Ndc80 complex's interaction with Dam1 complex. Average residence time of Ndc80-GFP mutant and wild-type complexes on microtubules in the presence or absence of Dam1 complex. Bars represent average residence time ± error of the mean (estimated by bootstrapping analysis; see Materials and methods for additional details). Ndc80-GFP complex microtubule residence time raw data are included in **Figure 1—source data 1**. Refer to **Supplementary file 1A** for statistical analysis of data in part (**D**).

The following source data and figure supplements are available for figure 1:

**Source data 1.** Table of Ndc80-GFP microtubule residence times for **Figure 1D**.

*Figure 1 continued on next page*

*Figure 1 continued*

**Figure supplement 1.** Dam1 and Ndc80 complexes robustly react with DSS and EDC cross-linking agents.

**Figure supplement 2.** Dam1p, Ask1p, and Spc34p cross-link to Nuf2p, in agreement to the coiled-coil structure of Ndc80p and Nuf2p.

**Figure supplement 3.** Ndc80 mutations outside of regions A, B, and C do not disrupt the interaction between Dam1 and Ndc80 complexes.

**Figure supplement 3—source data 1.** Table of Ndc80-GFP microtubule residence times for *Figure 1—figure supplement 3C*.

presence of taxol-stabilized microtubules. The C-terminal regions of Dam1p, Ask1p, and Spc34p formed cross-links to three distinct regions of Ndc80p (*Figure 1A*; *Figure 1—figure supplement 1*). The corresponding regions of Nuf2p also formed cross-links to these same regions of the Dam1 complex (*Figure 1—figure supplement 2*). These three C-terminal regions of the Dam1 complex include five sites phosphorylated by Aurora B kinase. Phosphorylation of all five sites fully disrupts the interaction between the Dam1 and Ndc80 complexes (*Lampert et al., 2010*; *Tien et al., 2010*). Thus, these sites likely represent primary interaction sites through which the Dam1 complex binds the Ndc80 complex.

To identify the cognate binding site(s) on the Ndc80 complex, we compared the cross-linking data to a map of functional regions on Ndc80p identified in a previous linker-scanning mutagenesis screen (*Tien et al., 2013*). Three of these regions map near or within the regions that cross-link to Dam1p, Ask1p, and Spc34p (*Figure 1A*). We define these interactions as A, B, and C, and we will refer to the specific regions on each protein that are involved in these interactions as $A^{Dam1p}$ and $A^{Ndc80p}$, $B^{Ask1p}$ and $B^{Ndc80p}$, and $C^{Spc34p}$ and $C^{Ndc80p}$, respectively (*Figure 1A* and *Table 1*).

We tested if the lethal mutations identified in our screen in regions $A^{Ndc80p}$, $B^{Ndc80p}$, and $C^{Ndc80p}$ (*Figure 1B*) interfere with interactions between the Ndc80 and Dam1 complexes on microtubules using single-molecule total internal fluorescence (TIRF) microscopy (*Figure 1C*). Importantly, Ndc80 complexes carrying a mutation in regions $A^{Ndc80p}$, $B^{Ndc80p}$, or $C^{Ndc80p}$ demonstrated homogeneous behavior in size exclusion chromatography, similar to the wild type Ndc80 complex (*Figure 1—figure supplement 3E*). These mutations also did not alter the residence time of the Ndc80 complex alone on microtubules, consistent with previous observations (*Tien et al., 2013*). As reported previously, the presence of the Dam1 complex significantly increases the residence time of wild-type Ndc80 complex on microtubules (*Tien et al., 2010*). However, addition of the Dam1 complex only partially increased the residence time of the mutant complexes as compared to the wild-type Ndc80 complex (*Figure 1D* and *Figure 1—figure supplement 3A,B*). Ndc80-GFP complex containing Ndc80p insertional mutations outside the regions $A^{Ndc80p}$, $B^{Ndc80p}$, and $C^{Ndc80p}$ had similar microtubule and Dam1 complex binding characteristics as the wild-type Ndc80 complex (*Figure 1—figure supplement 3C,D*). These observations suggest regions $A^{Ndc80p}$, $B^{Ndc80p}$, and $C^{Ndc80p}$ each contribute to the interaction of Ndc80 complex with Dam1 complex; and the insertional mutants in these

**Table 1.** The interacting regions in the Dam1 and Ndc80 complexes.

| Interaction | Protein region | Amino acids | Phosphorylated residues | Five amino acid mutation (insertion) position |
|---|---|---|---|---|
| A | $A^{Dam1p}$ | 241–330 | S257, S265, S292 | n/a |
| | $A^{Ndc80p}$ | 262–322 | n/a | 314 |
| B | $B^{Ask1p}$ | 133–225 | S200 | n/a |
| | $B^{Ndc80p}$ | 350–448 | n/a | 383 |
| C | $C^{Spc34p}$ | 118–274 | T199 | n/a |
| | $C^{Ndc80p}$ | 532–630 | n/a | 563 |

three regions were previously identified as lethal because they disrupted the interaction between the Dam1 and Ndc80 complexes.

## Dam1p, Ask1p, and Spc34p participate in interactions with Ndc80 complex that can be separately regulated by Aurora B kinase

We then asked if the corresponding regions in the Dam1 complex also contribute to the interaction between Dam1 and Ndc80 complexes. Aurora B kinase phosphorylates the Dam1 complex at six different sites: Dam1p S20, S257, S265, S292; Ask1p S200; Spc34p T199 (*Cheeseman et al., 2002*). Phosphorylation of Dam1p S20 significantly decreases Dam1 complex oligomerization, while phosphorylation of the three C-terminal sites in Dam1p slightly inhibits direct binding of the Dam1 complex to microtubules (*Gestaut et al., 2008*; *Zelter et al., 2015*). In our prior work, we showed that phosphorylating all five sites besides S20 fully disrupts interaction between the Dam1 and Ndc80 complexes (*Tien et al., 2010*). As shown in *Figure 1A*, these five sites fall into the three interaction regions defined above, A$^{Dam1p}$, B$^{Ask1p}$, and C$^{Spc34p}$ (*Table 1*). We therefore systematically phosphorylated combinations of these five sites to dissect their contributions to the interaction between the Dam1 and Ndc80 complexes.

We first purified recombinant mutant Dam1 complexes containing different combinations of Ser/Thr to Ala mutations, together with the S20A mutation in all cases. These mutants were treated with Aurora B kinase to produce a series of Dam1 complexes, each with a unique subset of phosphorylated sites. Each phosphorylated Dam1 complex was then tested for its ability to bind the wild-type Ndc80 complex in the single-molecule TIRF assay.

Simultaneously phosphorylating all three regions, A$^{Dam1p}$, B$^{Ask1p}$, and C$^{Spc34p}$ completely disrupted the interaction between the Dam1 and Ndc80 complexes, as reported previously (*Figure 2A* and *Figure 2—figure supplement 1A*; *Tien et al., 2010*). Individually phosphorylating regions A$^{Dam1p}$, B$^{Ask1p}$, or C$^{Spc34p}$ caused only partial disruption. Phosphorylating region B$^{Ask1p}$ together with C$^{Spc34p}$ also only partially disrupted the interaction, but phosphorylating A$^{Dam1p}$ in combination with either B$^{Ask1p}$ or C$^{Spc34p}$ was sufficient to fully disrupt the interaction (*Figure 2B* and *Figure 2—figure supplement 1B,C*). Neither the alanine mutations themselves, nor phosphorylation of a mutant Dam1 complex with all six sites mutated to alanine had any effect, confirming that the observed disruptions were due to phosphorylation at the known sites (*Figure 2A* and *Figure 2—figure supplements 1A* and *2*). Together these data demonstrate that each of these regions contributes to the interaction between the Dam1 and Ndc80 complexes and is regulated by phosphorylation.

## The Dam1 and Ndc80 complexes interact at three different sites

We next mixed and matched phosphorylation of the Dam1 complex with the mutations in Ndc80p. Full disruption by phosphorylation of the Dam1 complex was recapitulated by substituting a Dam1 complex phosphorylation with its corresponding mutation in Ndc80p. For example, phosphorylation of regions A$^{Dam1p}$ and C$^{Spc34p}$ fully disrupted the interaction between the two complexes (*Figure 2B*). Similarly, phosphorylation at region A$^{Dam1p}$ plus mutation in region C$^{Ndc80p}$ fully disrupted the interaction between the two complexes (*Figure 2C* and *Figure 2—figure supplement 1E,F*).

Phosphorylation combinations that partially disrupted the interaction between the Dam1 and Ndc80 complexes were also recapitulated by substituting a phosphorylation event with mutation in the corresponding binding region of Ndc80p. Phosphorylation at regions B$^{Ask1p}$ and C$^{Spc34p}$ partially disrupted the interaction between the two complexes (*Figure 2B*). Similarly, mutation in region B$^{Ndc80p}$ plus phosphorylation at region C$^{Spc34p}$, or phosphorylation at region B$^{Ask1p}$ plus mutation in C$^{Ndc80p}$ partially disrupted the interaction between the two complexes (*Figure 2C* and *Figure 2—figure supplement 1D*). Together, our results demonstrate that the Dam1 and Ndc80 complexes interact at three different sites, each of which is phospho-regulated by Aurora B kinase.

## The Ndc80 complex binds two Dam1 complex rings

The Ndc80 complex is predicted to be a long fibril. The coiled-coiled length between regions A$^{Ndc80p}$ and C$^{Ndc80p}$ is predicted to be 29 nm (*Lupas and Gruber, 2005*) (*Figure 3D*). The width of a Dam1 complex ring is only ~7 nm (*Ramey et al., 2011*). Assuming that the Ndc80 complex adopts

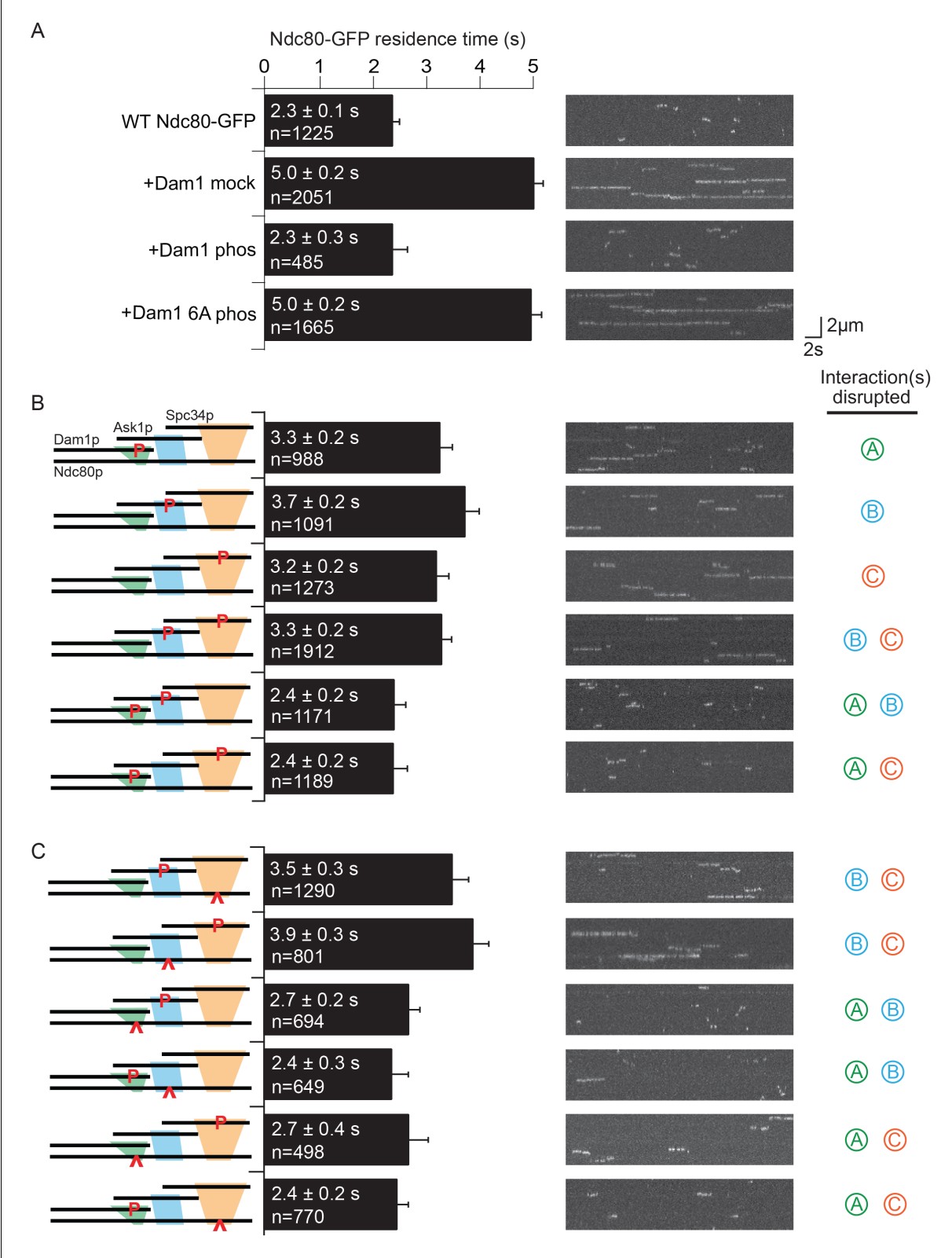

**Figure 2.** Dam1 and Ndc80 complexes interact through three distinct sites. (**A**) Average microtubule residence time of the wild-type Ndc80-GFP complex alone, in the presence of mock treated, phosphorylated Dam1 complex, or phosphorylated Dam1 6A mutant complex. Dam1 6A: all six Dam1 complex phosphorylation sites mutated to Ala. (**B**) Average microtubule residence time of wild-type Ndc80-GFP complex in the presence of selectively phosphorylated Dam1 complex. Diagram on the left indicates phosphorylated proteins. (**C**) Average microtubule residence time of Ndc80-GFP
*Figure 2 continued on next page*

*Figure 2 continued*

complex with lethal mutation in region $A^{Ndc80p}$, $B^{Ndc80p}$, or $C^{Ndc80p}$ in the presence of selectively phosphorylated Dam1 complex. Diagrams on the left indicate phosphorylated protein (P) and Ndc80p region with a lethal mutation (^). On the right are representative kymographs for each experiment. Bars represent average residence time ± error of the mean (estimated by bootstrapping analysis; see Materials and methods or additional details). Ndc80-GFP complex microtubule residence time raw data are included in *Figure 2—source data 1*. Refer to *Supplementary file 1C* for statistical analysis.

The following source data and figure supplements are available for figure 2:

**Source data 1.** Table of Ndc80-GFP microtubule residence times for *Figure 2*.

**Figure supplement 1.** Microtubule residence time survival probability plots of experiments in *Figure 2*.

**Figure supplement 2.** Mutating Dam1 complex Aurora B kinase phosphorylation sites to Ala does not affect the interaction between Dam1 and Ndc80 complexes.

**Figure supplement 2—source data 1.** Table of Ndc80-GFP microtubule residence times for *Figure 2—figure supplement 2*.

---

a conformation roughly parallel to the microtubule axis (*Alushin et al., 2010*; *Joglekar et al., 2009*), a single ring could not span this distance. Thus we considered that multiple Dam1 complex rings might bind to the Ndc80 complex. Using negative stain electron microscopy, we imaged the Dam1 complex on microtubules in the absence and presence of the wild-type Ndc80 complex.

The Dam1 complex formed rings on microtubules as previously reported (*Figure 3A,B*; *Miranda et al., 2005*; *Westermann et al., 2005*). In the absence of the Ndc80 complex, the Dam1 complex rings tended to bind in doublets of rings, at an inter-ring distance of 13.3 ± 2.4 nm (avg ± s.d.) (*Figure 3A–C* and *Figure 3—figure supplement 1A*). This suggests that Dam1 complex rings have an affinity for each other along the longitudinal axis of microtubules and have a tendency to stack together. The stacked double-ring organization may be analogous to the paired ring helices previously reported to form at higher concentrations of Dam1 complex (*Miranda et al., 2005*). The percent of Dam1 ring pairs found in closely spaced doublets was higher in the presence of the Ndc80 complex, 49.1% (estimated by Gaussian fitting to the peak, see Materials and methods), as compared to without the Ndc80 complex, 18.8%. In addition, the two rings in each doublet were consistently held apart by a distance of 33.1 ± 3.8 nm, similar to the predicted coiled-coil length between regions $A^{Ndc80p}$ and $C^{Ndc80p}$ (*Figure 3* and *Figure 3—figure supplement 1B*). This suggests that one Ndc80 complex bridges two Dam1 rings by binding one ring at region $A^{Ndc80p}$ and a second ring at region $C^{Ndc80p}$.

To further test this hypothesis, 10 coiled-coil heptad repeats were inserted into the Ndc80p and Nuf2p coiled-coil domains to increase the length of the coiled-coil domain between the interaction regions by a predicted 10.5 nm (*Figure 3D*). In the presence of the 10-heptad mutant Ndc80 (Ndc80$^{10\text{-hep}}$) complex, Dam1 ring doublets were consistently separated by 42.1 ± 2.1 nm, 9 nm farther apart than in the presence of wild-type Ndc80 complex (*Figure 3* and *Figure 3—figure supplement 1C*). Finally, mutation in regions A, B or C of Ndc80p abolished its ability to bind two rings as the Dam1 inter-ring distance in these cases was the same in the absence of Ndc80 complex (*Figure 3—figure supplements 1D–F and* and *2*). These results strongly support the hypothesis that the Ndc80 complex bridges two Dam1 complex rings in vitro.

## The Ndc80 10-heptad insertion complex is lethal in vivo

We next explored the possibility that Ndc80 complex might also bridge across two Dam1 complex rings in vivo. In particular, our observations raise the question of whether the specific 33 nm inter-ring distance, delimited by the Ndc80 complex, is important. To test this we constructed yeast expression plasmids for *ndc80$^{10\text{-hep}}$* and *nuf2$^{10\text{-hep}}$*, encoding Ndc80p and Nuf2p each with 10-heptad repeats inserted. Using a plasmid shuffle assay, we found that even when introduced together, *ndc80$^{10hep}$* and *nuf2$^{10hep}$* could not support growth, even though the encoded proteins were

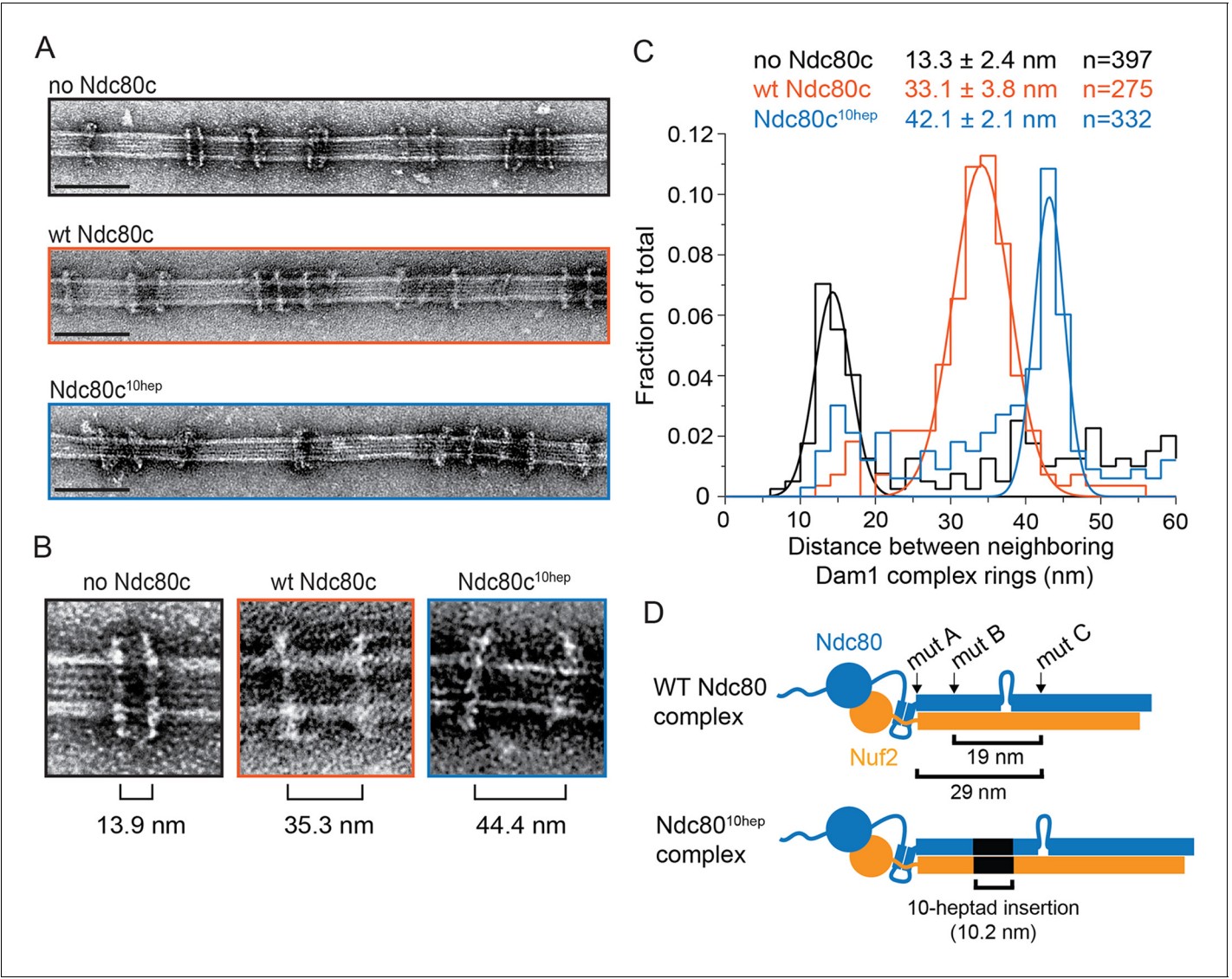

**Figure 3.** The Ndc80 complex binds two Dam1 complex rings. (A) Representative EM images of Dam1 complex rings on microtubules in the absence or presence of the Ndc80 complex. Three experiments include no Ndc80 complex (Ndc80c), wild-type Ndc80c, and Ndc80c[10hep]. Protein concentrations were 20 nM tubulin and 25 nM Dam1 and Ndc80 complexes. Scale bars: 100 nm. (B) Zoomed in images of a Dam1 complex ring doublet for the three experiments. Distances from middle of one ring to that of next ring are indicated. (C) Distribution of closest neighboring inter-ring distances measured for the three different conditions. Measurements made between 0–60 nm are shown. The cluster of distance measurements was fitted with a Gaussian distribution and the vertices ± standard deviations for each fit are listed. The raw data of Dam1 complex inter-ring distance measurements are included in *Figure 3—source data 1*. (D) Diagram demonstrating the predicted coiled-coil distances between the Ndc80p lethal mutations. Diagram below shows the location of additional 10-heptad insertion between the hypothesized binding sites of two Dam1 complex rings. Refer to *Supplementary file 1E* for statistical analysis of data in part (C).

The following source data and figure supplements are available for figure 3:

**Source data 1.** Table of Dam1 complex inter-ring distance measurements for *Figure 3C*.

**Figure supplement 1.** Full distributions of Dam1 complex inter-ring measurements for various experiments shown in *Figure 3* and *Figure 3—figure supplement 2*.

**Figure supplement 2.** Mutations in regions A[Ndc80p], B[Ndc80p], or C[Ndc80p] disrupts the Ndc80 complex's ability to bind two Dam1 complex rings.

**Figure supplement 2—source data 1.** Table of Dam1 complex inter-ring distance measurements for *Figure 3—figure supplement 2C*.

expressed (*Figure 4A,B*). The Ndc80[10hep] complex is competent to assemble in vitro, to bind micro-tubules, to interact with the Dam1 complex on microtubules, and to bind two Dam1 complex rings (*Figure 3*; *Figure 4C*). The only known difference is that the inter-ring distance is increased by 9 nm. These results suggest that the Ndc80[10hep] complex cannot support viability due to an altered Dam1 complex inter-ring distance.

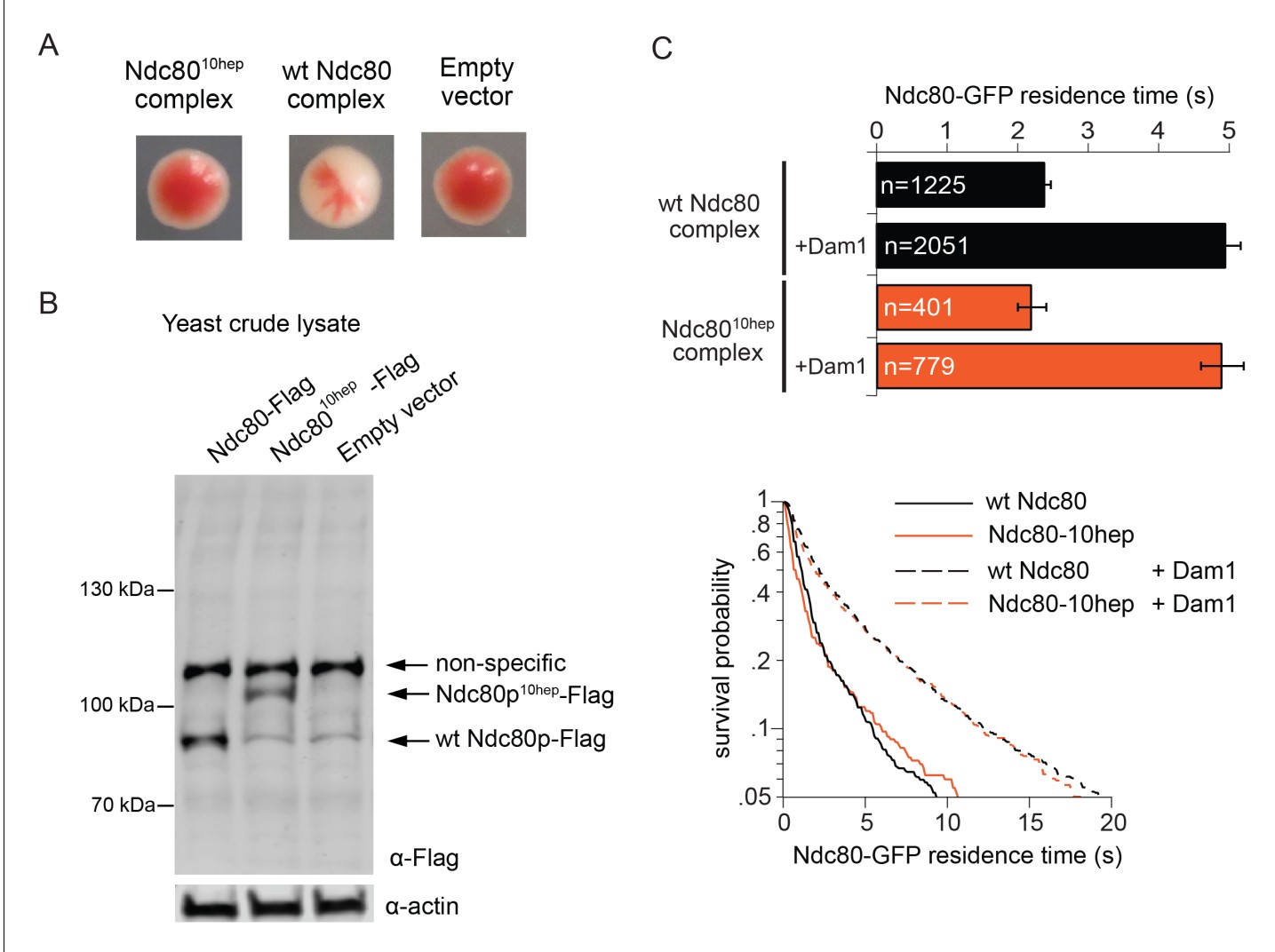

**Figure 4.** The Ndc80[10hep] complex does not support growth. (A) Red/white plasmid shuffle assay to test the viability of the Ndc80[10hep] complex. Solid red colonies indicate the inability of the Ndc80[10hep] complex and empty vector to support growth. Sectoring white colony indicates the ability of the wild-type Ndc80 complex to support growth. (B) α-FLAG immunoblot of crude lysate of cells expressing either wild-type Ndc80p, Ndc80p[10hep], or empty vector. α-actin immunoblot of the same lysates shows equal loading of the lysates. (C) Wild-type or Ndc80[10hep] complex average residence time (top) on microtubules in the absence and presence of Dam1 complex as derived from the survival probability curves (bottom). Bars represent average residence time ± error of the mean (estimated by bootstrapping analysis; see Materials and methods or additional details). Ndc80-GFP complex microtubule residence times raw data are included in *Figure 4—source data 1*. Refer to *Supplementary file 1F* for statistical analysis of data.

The following source data is available for figure 4:

**Source data 1.** Table of Ndc80-GFP microtubule residence times for *Figure 4C*.

## The Ndc80 mutants exhibit biorientation and attachment defects

The Ndc80p mutants in regions A[Ndc80p], B[Ndc80p], and C[Ndc80p] cannot bind two Dam1 complex rings, and consequently have decreased interaction with the Dam1 complex on microtubules. To determine if these Ndc80p mutants confer defects in chromosome biorientation and attachment, we used an auxin-inducible degron system for degrading the wild-type Ndc80-AID (*Nishimura et al., 2009*). Non-degradable wild-type or mutant *NDC80* was integrated at the *URA3* locus. Ndc80-AID was efficiently degraded in the presence of auxin (*Figure 5—figure supplement 1*). Budding index analyses revealed that cells carrying wild-type *NDC80* progressed through mitosis as expected. Cells without *NDC80* also budded and divided on schedule, despite major defects in kinetochore attachment (see below), as expected since activation of the SAC requires the Ndc80 complex (*Dou et al., 2015*; *Hiruma et al., 2015*; *Ji et al., 2015*). In contrast, cells carrying Ndc80p mutations in regions A[Ndc80p], B[Ndc80p], and C[Ndc80p] arrested as large-budded cells (*Figure 5—figure supplement 2*) suggesting defects in kinetochore attachment and indicating that the checkpoint remained functional.

We performed a detailed analysis of the phenotype conferred by the *NDC80* mutant alleles by imaging wild-type, depleted, and mutant cells with *CEN3* and spindle pole bodies (SPB) tagged with GFP and mCherry, respectively. Cultures were synchronized at G1, released from synchrony and then imaged after 60 min. In the majority of wild-type *NDC80* cells, CEN3 was bioriented with both CEN3-GFP puncta on the spindle axis; 78% were in metaphase and 22% were in anaphase as judged by the spindle length. In the majority of the Ndc80-depleted cells, CEN3-GFP was off of the spindle axis, consistent with complete detachment. A mutation in region A[Ndc80p], B[Ndc80p], or C[Ndc80p], resulted in 61, 59, and 75% of the cells with monooriented CEN3 and 34, 39, and 21% of cells with detached CEN3, respectively. (*Figure 5A,B*). In addition, in the majority of cells depleted of Ndc80 or with a mutation in regions A[Ndc80p], B[Ndc80p], or C[Ndc80p] the SPBs were separated by more than 2 μm (*Figure 5C*). These long spindles and the high frequency of detached CEN3 suggest that few if any of the kinetochores were bioriented. Thus, we conclude that Ndc80p carrying a mutation in region A[Ndc80p], B[Ndc80p], or C[Ndc80p] disrupts kinetochore biorientation and attachment in vivo.

## The Ndc80 mutants have defects in recruiting the Dam1 complex to the kinetochore

The Ndc80 complex is responsible for recruiting the Dam1 complex to the kinetochore (*Janke et al., 2002*). Since Ndc80 complexes carrying a mutation in region A[Ndc80p], B[Ndc80p], or C[Ndc80p] are defective in binding the Dam1 complex in vitro, we asked if Dam1 complex kinetochore localization is also compromised in vivo. We again used the auxin-inducible degron system for degrading the wild-type Ndc80-AID, while wild-type or mutant *NDC80* was integrated at the *URA3* locus; Mtw1-YFP, Dad4-CFP, and Spc110-mCherry marked the kinetochore, Dam1 complex, and SPB, respectively.

Cells expressing wild-type *NDC80* had two distinct clusters of Mtw1-YFP between two SPBs, as expected for bioriented sister kinetochores. Consistent with previous studies (*Cheeseman et al., 2001*; *Hofmann et al., 1998*; *Li et al., 2002*), the Dam1 complex localized along the spindle but also showed two distinct clusters that colocalized with the Mtw1-YFP. (*Figure 6A,B*). None of the mutants showed the organized kinetochores seen in wild-type cells. Instead, in the milder mutants (mutation in region B[Ndc80p]; C[Ndc80p]; or B[Ndc80p] and C[Ndc80p]) kinetochores were spread along the spindle and a few were detached. In the more severe mutants (mutation in region A[Ndc80p]; A[Ndc80p] and B[Ndc80p]; A[Ndc80p] and C[Ndc80p]; A[Ndc80p], B[Ndc80p] and C[Ndc80p]; or Ndc80-depleted) few kinetochores were associated with the spindle. Instead, most were detached from the microtubules between the two SPBs (*Figure 6—figure supplement 1*).

Unlike wild-type *NDC80*, Dad4-CFP in mutant Ndc80 cells decorated the spindle but concentrated poorly at the Mtw1-YFP clusters that did occur (*Figure 6B* and *Figure 6—figure supplement 1*). In wild-type *NDC80* cells, an increase or decrease in Mtw1-YFP fluorescence correlated with an increase or decrease in Dad4-CFP fluorescence, respectively; however, Ndc80-depleted cells showed poor correlation. One or multiple mutations in region A[Ndc80p], B[Ndc80p], and C[Ndc80p] resulted in significantly decreased correlation between Mtw1-YFP and Dad4-CFP (*Figure 6C*). Single mutations in region B[Ndc80p] or C[Ndc80p] resulted in less severe defects. These observations suggest that mutations in regions A[Ndc80p], B[Ndc80p], and C[Ndc80p] had little effect on binding of Dam1 complex to the microtubules, but disrupted the ability of Ndc80 to recruit Dam1 complex to the kinetochore in vivo, thereby causing severe defects in kinetochore attachment and biorientation.

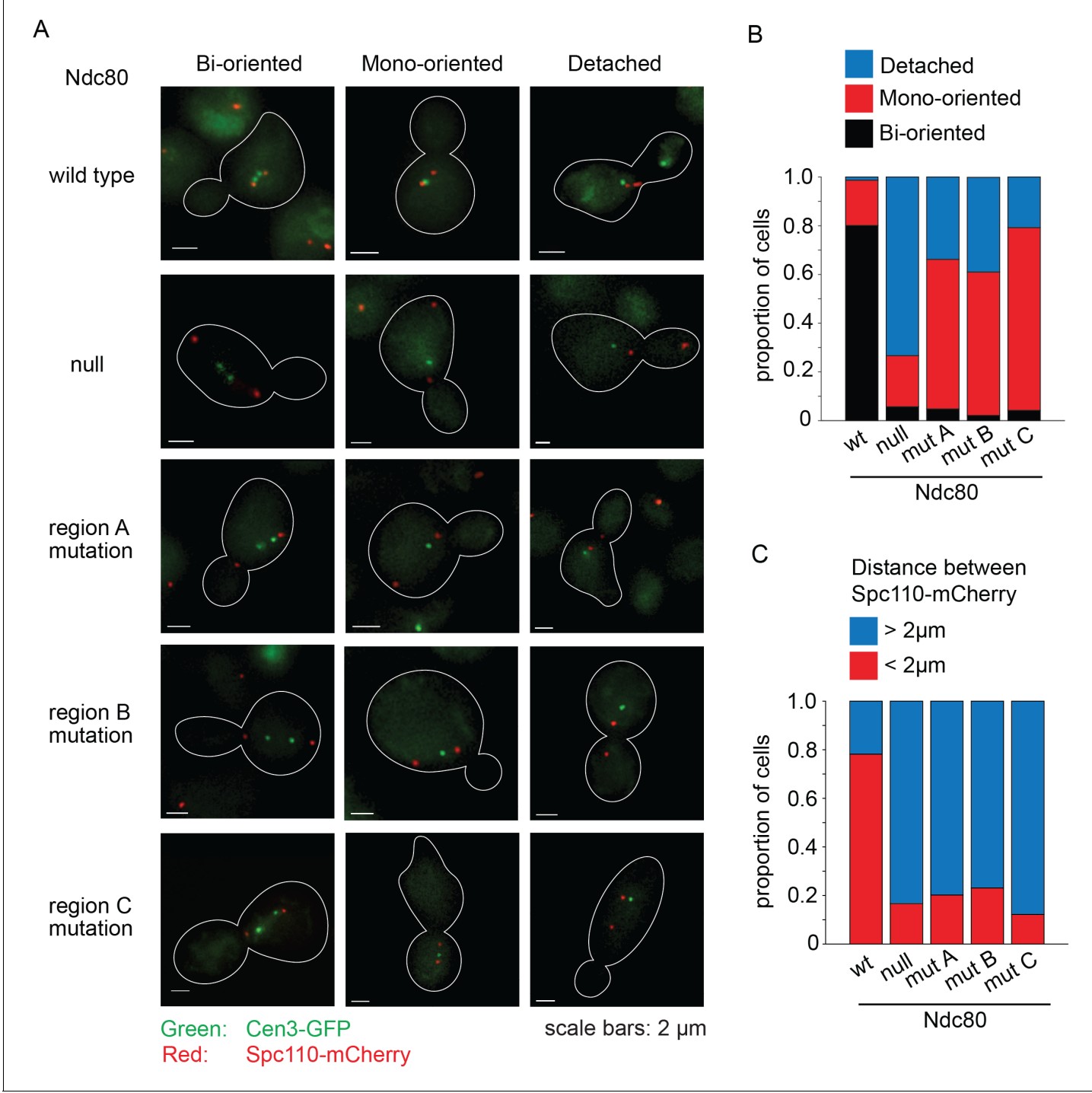

Green: Cen3-GFP
Red: Spc110-mCherry

scale bars: 2 µm

**Figure 5.** Lethal mutations in A[Ndc80p], B[Ndc80p], and C[Ndc80p] have biorientation and microtubule attachment defects in vivo. (**A**) Representative images from Ndc80-AID degron experiments with cells carrying an extra copy of *NDC80* wild-type, no *NDC80* (null), or a mutations in A[Ndc80p], B[Ndc80p], or C[Ndc80p]. Cells were treated with 6 µM α-factor for two hours prior to treatment with 6 µM α-factor and 0.5 mM auxin for one additional hour. At time 0, cells were released from α-factor and incubated in YPD medium containing 0.5 mM auxin. Images were taken at 60 min after release from α-factor arrest. Representative images are shown. (**B**) Stacked bar graphs showing proportion of the cells containing bioriented, monooriented or detached CEN3-GFP. Spindles with two CEN3-GFP puncta positioned between the two SPBs were counted as bioriented. Spindles with one CEN3-GFP puncta positioned between the two SPBs were counted as monooriented and spindles with one (or rarely two) CEN3-GFP puncta positioned off the spindle axis were counted as detached. Only cells with two Spc110-mCherry puncta and at least one CEN3-GFP punctum were included in the analysis. (**C**)

*Figure 5 continued on next page*

*Figure 5 continued*

Stacked bar graph showing the distance between SPBs. Wild-type *NDC80* n = 236 cells; no *NDC80* n = 210 cells; A$^{Ndc80p}$n = 210 cells; B$^{Ndc80p}$n = 190 cells; C$^{Ndc80p}$n = 260 cells.

The following figure supplements are available for figure 5:

**Figure supplement 1.** Wild-type Ndc80-AID is degraded upon the addition of auxin.

**Figure supplement 2.** Lethal mutation in region A$^{Ndc80p}$, B$^{Ndc80p}$, or C$^{Ndc80p}$ causes mitotic arrest.

## Discussion

The discovery that the Dam1 complex forms oligomers and rings around microtubules galvanized the mitosis field by providing a molecular explanation for how kinetochores are able to maintain processive attachment to the flared ends of a depolymerizing microtubule. Rings are not required for the Dam1 complex to track depolymerizing microtubules in the absence of tension (*Gestaut et al., 2008*; *Grishchuk et al., 2008*). However, oligomerization of the Dam1 complex is required to maintain attachments under tension as would be experienced during metaphase (*Umbreit et al., 2014*; *Volkov et al., 2013*). Prior models of the kinetochore have all assumed that each kinetochore contains only one Dam1 complex ring. Here we show that the Ndc80 complex bridges two rings of the Dam1 complex in vitro. Lethal mutations in Ndc80p that block binding of either one of the two rings result in loss of the Dam1 complex from the kinetochore and failure in biorientation and attachment of kinetochores to the mitotic spindle. These results suggest that faithful microtubule attachments require two Dam1 complex rings per kinetochore in vivo.

Counting of kinetochore components by fluorescence microscopy has yielded conflicting results for the number of Dam1 complexes at the yeast kinetochore depending on the standard used. Assuming two Cse4 histone molecules per kinetochore, Joglekar and coworkers report enough Dam1 complexes to form one ring at a kinetochore during metaphase and a partial ring during anaphase (*Aravamudhan et al., 2013*; *Joglekar et al., 2006*). Lawrimore and coworkers measured 5.5 Cse4 histone molecules per kinetochore, giving enough Dam1 complex for two rings at a kinetochore during anaphase (*Lawrimore et al., 2011*); whereas Coffman and coworkers measured 8 Cse4 histone molecules per kinetochore (*Coffman et al., 2011*). Our results show that the Ndc80 complex can bind two Dam1 rings in vitro. Given the results of the latter two groups, we propose that there are two Dam1 complex rings at the kinetochore.

The region before the hairpin and the loop region of Ndc80p have both been identified as binding sites for the Dam1 complex (*Lampert et al., 2013*; *Maure et al., 2011*). Our results indicate that the interaction between the Ndc80 and Dam1 complexes is more extensive forming a tripartite network. The C-terminal domains of Dam1p, Ask1p, and Spc34p interact with three distinct regions of Ndc80p, one of which is the hairpin consistent with the previous report (*Lampert et al., 2013*). Our results are also consistent with a previous study showing an interaction between Ndc80p and the C-terminal region of Dam1p (*Kalantzaki et al., 2015*). However, we did not find any evidence for an interaction between the Ndc80p 'loop' region and Dam1p (*Maure et al., 2011*). The inability of the loop deletion mutant to assemble the Dam1 complex into the kinetochore might be due to an indirect effect.

Comparing the predicted distances between regions A$^{Ndc80p}$, B$^{Ndc80p}$, and C$^{Ndc80p}$, and the 33 nm Dam1 complex inter-ring distance from EM experiments suggests that regions A and B contribute to the Ndc80 complex interaction with the ring closer to the N-terminus of Ndc80p/Nuf2p, while region C contributes to the Ndc80 complex interaction with the second ring (*Figure 7*). Our TIRF microscopy and live cell imaging results suggest that disrupting interaction A is more severe than disrupting interaction B or C. A recent computational model of the structure of the Dam1 complex bound to microtubules positions region A$^{Dam1p}$ and B$^{Ask1p}$ near the microtubule surface, while region C$^{Spc34p}$ is positioned away from the microtubule surface (*Zelter et al., 2015*). Accordingly, our model of the interaction between Dam1 and Ndc80 complexes shows the Ndc80 complex positioned under the first Dam1 ring and above the second Dam1 complex ring further from Ndc80p N-terminus (*Figure 7*).

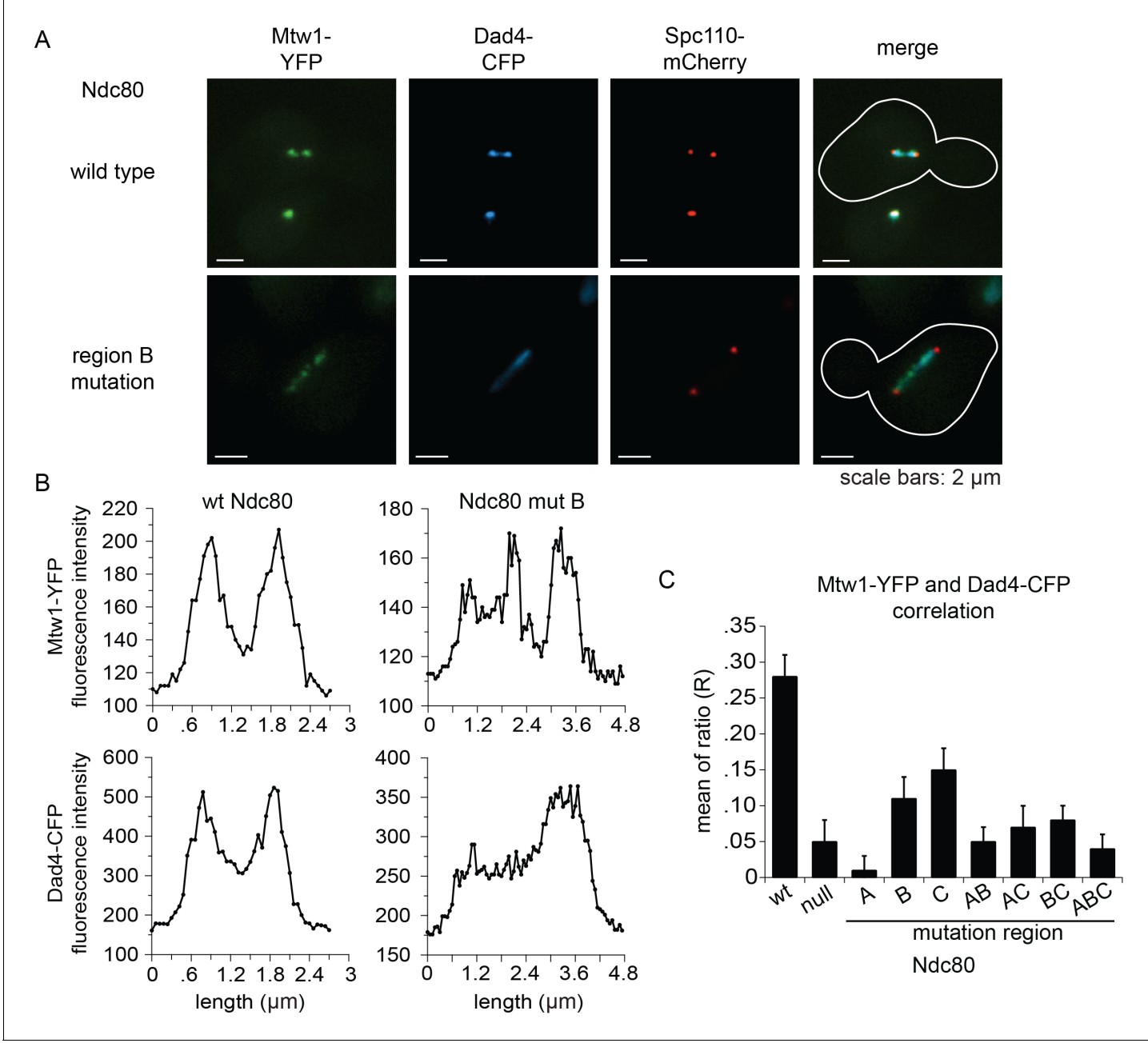

scale bars: 2 µm

**Figure 6.** Lethal mutations in A$^{Ndc80p}$, B$^{Ndc80p}$, and C$^{Ndc80p}$ confer defects in Dam1 complex recruitment to the kinetochore. (**A**) Representative images from *NDC80-AID* degron experiments with cells also containing wild-type *NDC80* or mutation in region B$^{Ndc80p}$. The inner kinetochore was visualized with Mtw1-YFP, the Dam1 complex with Dad4-CFP, and the SPB with Spc110-mCherry. Cells were treated with 6 µM α-factor for two hours prior to the addition of 0.5 mM auxin for one additional hour. At time 0, cells were released from α-factor and incubated in YPD medium containing 0.5 mM auxin. Images were taken at 60 min after release from α-factor arrest. (**B**) Line scan traces of Mtw1-YFP and Dad4-CFP from the images in (**A**). Only cells with two SPBs in the same plane of focus were selected for analysis. Each line scan extended from one SPB to the other. Each point represents one pixel. (**C**) Summary of correlation analysis carried out between Mtw1-YFP and Dad4-CFP. For each line scan, a correlation of the positive or negative changes of pixel intensity along the line between the Mtw1-YFP and Dad4-CFP channels was calculated to examine co-localization along the spindle (see Materials and methods). Between 18 and 33 cells were analyzed for each mutant. Mtw1-YFP and Dad4-CFP fluorescence intensity raw data are included in *Figure 6—source data 1*. Refer to **Supplementary file 1G** for statistical analysis of data.

The following source data and figure supplement are available for figure 6:

**Source data 1.** Tables of Mtw1-YFP and Dad4-CFP fluorescence intensities for cells analyzed in *Figure 6C*.

*Figure 6 continued on next page*

*Figure 6 continued*

**Figure supplement 1.** Mutations in regions A[Ndc80p], B[Ndc80p], or C[Ndc80p] results in severe defects in kinetochore biorientation and attachment.

Aberrant kinetochore-microtubule attachments are phosphorylated and destabilized by Aurora B kinase (*Biggins et al., 1999*; *Cheeseman et al., 2002*; *Hauf et al., 2003*; *Pinsky et al., 2006*; *Tanaka et al., 2002*). Aurora B kinase phosphorylation of the Dam1p, Ask1p, and Spc34p C-terminal sites together fully disrupts the interaction between the Dam1 and Ndc80 complexes (*Tien et al., 2010*). This regulation mechanism has been thought of as binary, either full or no disruption of the interaction.

We show that phosphorylation of certain combinations results in only partial disruption and full disruption does not require phosphorylation of all three proteins. Our results suggest how the Aurora B kinase might fine tune the interaction between Dam1 and Ndc80 complexes, resulting in a range of disruption.

The inability of *ndc80*[10hep] and *nuf2*[10hep] together to support growth in vivo might be explained in several ways. The addition of extra heptad repeats in the coiled-coil domains might destabilize formation of the heterotetrameric Ndc80 complex, for example. However, we do not currently favor this hypothesis because the extra-long mutant version of Ndc80p is expressed in vivo, and because the Ndc80[10hep] complex is stable and binds to both microtubules and to the Dam1 complex in vitro, similar to wild-type Ndc80 complex. Unnatural orientation of the Ndc80p/Nuf2p globular head in the Ndc80[10hep] complex, due to the addition of extra heptad repeats, might also disrupt the specific geometry that the Ndc80 complex takes in relation to other kinetochore proteins. However given

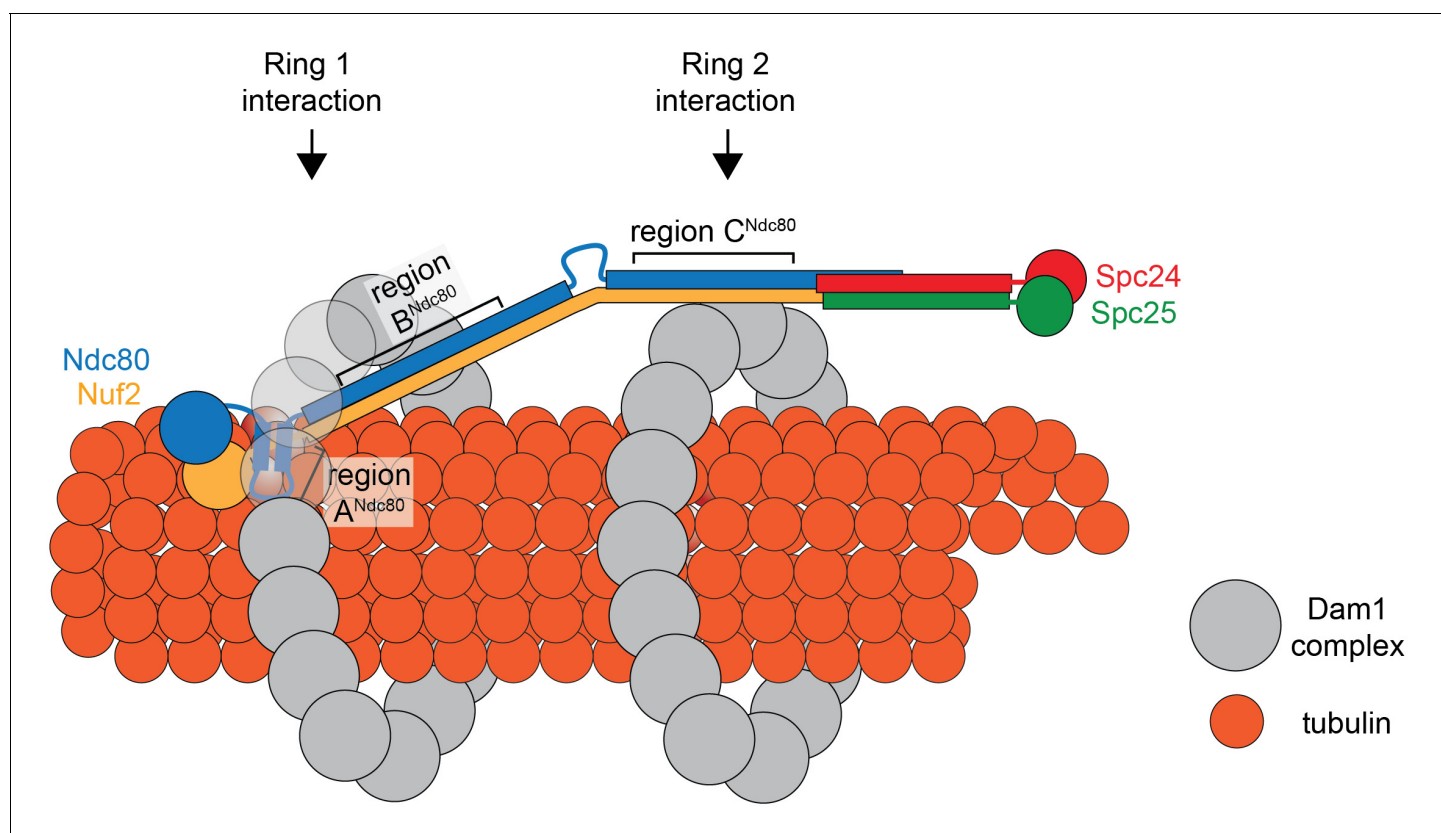

**Figure 7.** Model of the Ndc80 complex bridging two Dam1 complex rings, separated by 33 nm. Results in this study support ring 1 interaction being comprised of interaction A and B, and ring 2 interaction being comprised of interaction C (refer to *Figure 1*). This image depicts only one Ndc80 complex across two Dam1 complex rings, but multiple Ndc80 complexes would be present in vivo.

the presence of the flexible loop in Ndc80p, it seems unlikely that the two ends of the complex are held in a rigid orientation relative to each other. Compared to the wild-type Ndc80 complex, the Ndc80[10hep] complex further separates the Dam1 complex rings in vitro. We propose that the specific 33 nm Dam1 complex inter-ring distance is vital. The transition from 13 nm, in the absence of the Ndc80 complex, to 33 nm Dam1 complex inter-ring distance could signal the establishment of kinetochore-microtubule attachment or biorientation. Deviation from the specific 33 nm distance might disrupt a yet unknown mechanism for detecting kinetochore-microtubule attachments, or bioriented attachments, or both. Future work will focus on characterizing the specific effects of the extra-long Ndc80 complex and the role of the Dam1 complex inter-ring distance in vivo.

## Materials and methods

### Protein expression and purification

The *S. cerevisiae* Ndc80 and Dam1 complexes were independently expressed in *E. coli* using polycistronic vectors, as previously described (*Gestaut et al., 2008*; *Miranda et al., 2005*; *Powers et al., 2009*; *Tien et al., 2010*; *Wei et al., 2005*). The Dam1 complex component Spc34p C-terminus was tagged with Hisx6 and the Ndc80 complex component Spc24 N-terminus was tagged with FLAG or Hisx6. Each complex was affinity-purified before further purification through gel filtration. For TIRF microscopy, Nuf2p C-terminus of the Ndc80 complex was tagged with GFP.

### Dam1 complex phosphorylation

GST-Ipl1 and GST-Sli15 (residues 554–698) were purified as previously described (*Gestaut et al., 2008*; *Tien et al., 2010*; *Zelter et al., 2015*). GST-Ipl1 (pSB196, Sue Biggins, Fred Hutchinsin Cancer Research Center, Seattle, WA) and GST-Sli15 (residues 554–698) (pSB503, Sue Biggins) were expressed at 23°C and 37°C, respectively for 2 hr. GST-Ipl1 was purified using GSTrap HP (GE Healthcare Biosciences, Pittsburgh, PA) following manufacturer's instructions, except that the elution buffer was 50 mM Tris buffer (pH 8.0), 250 mM KCL, 10 mM glutathione. HiTrap 26/10 Desalting column (GE Healthcare) was used to exchange the buffer to 50 mM HEPES buffer (pH 7.4), 100 mM NaCl. GST-Sli15 was purified with glutathione-Sepharose 4B resin (GE Healthcare) following manufacturer's instructions, except that the elution buffer was 20 mM Tris buffer (pH 8.0), 200 mM NaCl, 1 mM $\beta$-mercaptoethanol, 1 mM EDTA, 10 mM glutathione.

4 µM recombinant Dam1 complex was incubated with 0.5 µM GST-Ipl1, 0.5 µM GST-Sli15, 200 mM NaCl, 10 mM ATP, 25 mM MgCl$_2$, and 50 mM HEPES buffer, pH 7.4. Reaction mixtures were incubated for 90 min at 30°C. Under these conditions, we achieve nearly stoichiometric phosphorylation of the complex (*Gestaut et al., 2008*). Mock treated (non-phosphorylated) controls of the Dam1 complex was carried out by substituting ATP with dH$_2$O.

### TIRF microscopy

Glass slides and functionalized coverslips were used to construct flow chambers, as reported previously (*Gestaut et al., 2008*, *2010*; *Tien et al., 2010*; *Zelter et al., 2015*). A coverslip was adhered to a glass slide with double-sided tape, forming a flow chamber between two adjacent strips of tape. 'Rigor' kinesin was flowed through the chamber to non-specifically bind to the coverslip. Taxol-stabilized, Alexa-647-labelled microtubules were then flowed in and incubated for 5 min for immobilization. For all TIRF microscopy experiments, Ndc80-GFP complex was incubated at 50 pM for single-molecule imaging. Each phosphorylated or mock-treated Dam1 complex mutant construct was incubated at 2.5 nM. GFP and Alexa-647 fluorescence channels were simultaneously recorded using a custom TIRF imaging system (*Gestaut et al., 2010*). All TIRF microscopy experiments were carried out in BRB80 (80 mM PIPES buffer (pH. 6.8), 1 mM EGTA, 1 mM MgCl$_2$) in the presence of 8 mg/ml BSA, 0.04 mg/ml κ-casein, and an oxygen scavenger system (200 µg/ml glucose oxidase, 35 µg/ml catalase, 25 mM glucose, and 5 mM DTT). Each experimental condition was assayed between three and seven times yielding between 401 and 2051 measurements. This sample size adequately identified possible differences between different conditions.

Analysis of the single particle tracking was carried out as previously described (*Tien et al., 2010*; *Umbreit et al., 2014*; *Zelter et al., 2015*). Custom analysis software was developed in Labview (National Instruments) (RRID: SCR_014325) (*Source code files 1–3*) and Igor Pro (Wavemetrics)

(RRID: SCR_000325) (*Source code files 4–6*). Bootstrapping analysis was used to calculate the mean residence times (*Umbreit et al., 2014*). Randomly resampling each dataset (residence times from each TIRF microscopy experimental condition) with replacement was repeated 1000 times. Carrying out this method with each dataset yielded normal distributions. Gaussian fits to these distributions were used to estimate the mean residence times and the width of the fit was used as an estimate of error. Statistical analysis of the TIRF data was performed with pairwise z-tests. P-values were computed form the pairwise z-scores: $z = (\mu_1 - \mu_2) \cdot (\delta_1^2 + \delta_2^2)^{-0.5}$. $\mu_1$ and $\mu_2$ are bootstrap average residence times of Ndc80-GFP complex. $\delta_1^2$ and $\delta_2^2$ are the corresponding estimates of error. Tables of pairwise comparisons can be found in *Supplementary file 1A-D* and *F*.

## Chemical Cross-linking and mass spectrometry analysis (XL-MS)

XL-MS experiment was performed twice, using EDC and DSS cross-linking agents (biological replicate). The two experiments identified similar interaction regions between the Dam1 and Ndc80 complexes. XL-MS was performed as described in *Zelter et al. (2015)*. Briefly, 10 µg of taxol stabilized microtubules were mixed with 10 µg Dam1 complex and 10 ug Ndc80 complex in 100 µL BRB80 at 25°C and allowed to stand for 5 mins. For DSS cross-linking, 3 µL 14.5 mM DSS in DMSO was added, and the mixture allowed to cross-link for 2 mins at 25°C before quenching by addition of 10 µL 0.5 M NH$_4$HCO$_3$. For EDC cross-linking 7.5 µL 145 mM EDC plus 3.75 µL 145 mM Sulfo-NHS were added to the reaction, and the mixture allowed to cross-link for 30 mins at 25°C before quenching by addition of 10 µL 0.5 M NH$_4$HCO$_3$. Quenched reactions were spun at 58,000 rpm in a TLA100 rotor for 10 min at 37°C. The pellet was resuspended in 100 µL ice cold buffer reduced with 10 mM dithiothreitol (DTT) at 37°C for 30 min followed by 30 mins alkylation at RT with 15 mM iodoacetamide (IAA). Digestion with trypsin at a substrate to enzyme ratio of 60:1 was performed overnight at room temperature with shaking. Digested samples were acidified with 5 M HCL prior to being stored at −80°C until analysis. Mass spectrometry and data analysis was performed on either a Q-Exactive or Q-Exactive HF (Thermo Fisher Scientific, Waltham, MA) as previously described (*Zelter et al., 2015*). Each cross-linked sample was run four times and the data were combined before analysis (technical replicate). Mass spectra were converted into mzML using msconvert from ProteoWizard (*Chambers et al., 2012*). Standard linear peptide searches were performed using Comet to identify all proteins in the sample (*Eng et al., 2013*). Cross-linked peptides were identified using the Kojak version 1.4.2 cross-link identification software (*Hoopmann et al., 2015*) following the author's instructions (http://www.kojak-ms.org). Kojak results were exported to Percolator version 2.08 (*Käll et al., 2007*) to produce a statistically validated set of cross-linked peptide spectrum matches (PSMs) at the desired false discovery rate (FDR) threshold. Percolator (RRID: SCR_005040) is a semi-supervised algorithm that assigns a statistically meaningful q value to each PSM through analysis of the target and decoy PSM distributions. Decoy PSMs derive from peptide sequences known to be false. In the current work the target databases consisted of all proteins identified in the sample analyzed, while the decoy databases consisted of the corresponding set of reversed protein sequences. Cross-link PSMs were considered to be false if at least one of the peptides was from a decoy protein sequence. The data presented in this paper was filtered to show only hits to the target proteins that had a Percolator assigned peptide level q value ≤ 0.05. The complete, unfiltered list of all PSMs and their Percolator assigned q values, are available on the ProXL web application (*Riffle et al., 2016*) at: http://proxl.yeastrc.org/proxl/viewProject.do?project_id=24 along with the raw MS spectra and search parameters used.

## Electron microscopy

Cleared tubulin was polymerized in a total volume of 40 µl BRB80 (80 mM PIPES buffer (pH 6.8), 1 mM EGTA, 1 mM MgCl$_2$) containing 1.75 mM GTP, 1 mM MgCl$_2$, and 3.5% DMSO at 37°C for 30 min. Microtubules were pelleted and resuspended in BRB80 containing 10 µM taxol. All samples were prepared in BRB80 +10 µM taxol by mixing 20 nM microtubules, and 25 nM Dam1 in the absence or presence of 25 nM Ndc80 complex. Samples were incubated at room temperature for 15 min. Carbon-coated copper grids were negatively discharged in a glow discharge device. A 5 µl volume of sample was applied on a discharged grid for 20 s before being blotted. 6 µl 2% uranyl acetate was then applied on the grid and incubated for 1 min. The grid was blotted and air dried. EM samples were viewed on a transmission electron microscope (Morgagni; FEI, Hillsboro, OR)

operating at 100 kV. Images were recorded on a bottom-mounted Orius (Gatan, Pleasanton, CA) camera at 22,000x magnification. ImageJ was used to measure the Dam1 complex inter-ring distances. Distances from middle of a ring to the middle of the next closest ring, on both sides, were measured.

The EM microscopy experiments were performed with two independent copper grids. Similarity between the two replicates allowed us to combine the data. Statistical analysis of the EM data was performed with pairwise z-tests. P-values were computed from the pairwise z-scores: $z = (\mu_1 - \mu_2) \bullet (\delta_1^2 + \delta_2^2)^{-0.5}$. $\mu_1$ and $\mu_2$ are the center of the Gaussian fits around the single large cluster in each sample and $\delta_1^2$ and $\delta_2^2$ are the corresponding estimates of error. A table of pairwise comparisons can be found in *Supplementary file 1E*.

## Yeast strain construction and validation

Strains used in this study are derivatives of W303 (*Supplementary file 2A*). We have previously verified the genotype of the W303 by whole-genome tiling as described (*Gresham et al., 2006*). For strains constructed for this study, *NDC80* alleles were integrated at the *URA3* locus as previously described (*Widlund and Davis, 2005*). Plasmids containing the *NDC80* alleles were sequenced before transformation. Successful integration was verified by PCR. Genes were tagged with genes encoding fluorescent proteins as described (*Wach et al., 1997*) and verified by PCR.

## Test for function of Ndc80$^{10hep}$ complex

To test if the Ndc80$^{10hep}$ complex is functional in vivo, we used a red/white plasmid shuffle assay (*Davis, 1992*; *Muller, 1996*) using strain JTY5-8B (*ade2-1oc ade3Δ−100 leu2-3,112 ura3-1 ndc80Δ:: natMX*), harboring pJT12 (*NDC80 ADE3* in a 2 μm vector). JTY5-8B was transformed with pJOK13 (*URA3 ndc80$^{10hep}$*) and selected for growth on SD-ura low adenine plates. Transformation with pJOK13 yielded non-sectoring red colonies (strain JOKY3) demonstrating that the *ndc80$^{10hep}$* is not viable when paired with wild-type *NUF2* (data now shown). JOKY3 was then grown in SD-lys-ura media before being transformed with pJOK018 (*LEU2 nuf2$^{10hep}$*), pJOK019 (*LEU2 NDC80*), or pRS315 (*LEU2*) and plated on SD-ura-leu plates. Four colonies from each transformation were streaked onto SD-ura low-ade plates to screen for sectoring colonies (technical replicate). Two transformations were performed for each condition (biological replicate), both trials gave the same result.

## Yeast live-cell imaging and analysis

Strains for live-cell imaging were constructed using previously described strains: *SPC110-mCherry*, *pCUP1-GFP-LacI12* and *CEN3::33LacO* (*Wargacki et al., 2010*); *NDC80-3V5-IAA7* and *pGPD1-TIR1* (*Miller et al., 2016*). Sue Biggins kindly provided the Ndc80-AID strain. Strains containing *NDC80*, *ndc80-314* (ins A$^{Ndc80p}$), *ndc80-383* (ins B$^{Ndc80p}$), or *ndc80-563* (ins C$^{Ndc80p}$) were constructed through an integration method previously described (*Widlund and Davis, 2005*). YFP and CFP were integrated into Mtw1 and Dad4, respectively, through homologous recombination with PCR products amplified with primers containing homologous sequence of its target locus.

Asynchronously growing cells (30 Klett units) were arrested with 6 μM α-factor for 2 hr. Auxin (IAA (0.5 mM) was then added and the incubation continued for an additional hour. The cells were released from α-factor into YPD +0.5 mM IAA medium and imaged at 15 min time-intervals from time 0 to 120 min. At each time point, cells were also fixed in formaldehyde for budding index analysis. Budding index was determined twice (from two different colonies) and yielded the same conclusion for each of the five yeast strains shown in *Figure 5*. Data is shown for one of the two experiments. Cells were placed on agar pads prior to imaging, as previously described (*Muller et al., 2005*). Cells were imaged using a DeltaVision system (Applied Precision, Issaquah, WA) equipped with IX70 inverted microscope (Olympus, Center Valley, PA), a Coolsnap HQ digital camera (Photometrics, Tucson, AZ), and a U Plan Apo 100X objective (1.35 NA).

For CEN3 tracking biorientation assay, *CEN3-LacI* was visualized with *GFP-LacO* and the spindle pole bodies were visualized with *SPC110-mCherry*. Both GFP and mCherry were imaged with 0.4 s exposures with 16, 0.2 μm z-sections. Images were binned 2 × 2 with a final resolution of 512 × 512. Biorientation analysis was done manually for those cells with 2 mCherry puncta using Imaris software (Bitplane, Switzerland) (RRID: SCR_007370). For all the strains tested, the 60 min time point was chosen for data analysis. The biorientation assay was performed twice for each strain, each time with a

different colony and on a different day. The combination of two experiments yielded sample sizes between 190 and 260 cells.

For Dam1 complex kinetochore localization assay, kinetochore, Dam1 complex, and spindle pole bodies were visualized with Mtw1-YFP, Dad4-CFP, and Spc110-mCherry respectively. All three proteins were imaged with single focal plane 0.4 s exposures. Images were not binned with a final resolution of 1024 × 1024. Line scans analyses were obtained of Mtw1-YFP and Dad4-CFP fluorescence using Imaris software (Bitplane) (RRID: SCR_007370) and extended from one SPB to the other. For each line scan, a correlation of the sign (positive or negative) of pixel intensity changes across the line between the Mtw1 and Dad4 channels was calculated to examine co-localization along the spindle. This was calculated as: $R = N_{agree} - N_{disagree} / N - 1$, where $N_{agree}$ is the number of pixels that changed with the same sign (both channels increased or decreased), $N_{disagree}$ is the number of pixels that changed with opposite signs (one channel increased and one decreased) and N is the length of the line scan. For each strain, a mean R was calculated from all line scans. The localization assay was performed twice for each strain, each time with a different colony and on a different day. The combination of two experiments yielded sample sizes between 18–33 cells. Statistical analysis of the Mtw1-YFP and Dad4-CFP mean R-scores was performed with Student's t-test. A table of the pairwise comparisons can be found in *Supplementary file 1G*.

## Western blot analysis

During the yeast live-cell imaging, samples were collected for western blot analysis. After the addition of auxin, cell samples were collected at various time points. 0.7 ml cells and 0.7 ml 20% trichloroacetic acid were incubated at 4℃ for 1 hr. Samples were pelleted for at 20,000xg for 10 min at 4℃. The pellet was washed by resuspending in 1 ml 4℃ ethanol. The sample was then pelleted and resuspended once more before a final pelleting step. Supernatant was discarded and sample was air dried before resuspending in 50 µl 0.1N NaOH, SDS-PAGE sample buffer. Western blot analysis was performed, probing for Ndc80p (gift from Arshad Desai, Ludwig Cancer Research Center, University of California San Diego) and $\beta$-Actin (Abcam, Cambridge, MA). Protein levels analysis was carried out using Image Studio Lite (Li-Cor, Lincoln, Nebraska) (RRID: SCR_014211). Each western blot was performed twice, yielding the same conclusion for each experiment.

## Acknowledgements

We thank Justin Kollman and Annie Dosey for kind guidance with EM experiments. Also, we thank Matthew Miller and Sue Biggins for gracious gifts of Ipl1-GST constructs and yeast strains. This work was funded by National Institutes of Health grants R01 GM040506 to TND, P41 GM103533 to MJM and R01 GM079373 and S10 RR026406 to CLA.

## Additional information

### Funding

| Funder | Grant reference number | Author |
| --- | --- | --- |
| National Institutes of Health | R01 GM040506 | Trisha N Davis |
| National Institutes of Health | P41 GM103533 | Michael J MacCoss |
| National Institutes of Health | R01 GM079373 | Charles L Asbury |
| National Institutes of Health | S10 RR026406 | Charles L Asbury |

The funders had no role in study design, data collection and interpretation, or the decision to submit the work for publication.

### Author contributions

JoK, Conceptualization, Data curation, Formal analysis, Validation, Investigation, Visualization, Methodology, Writing—original draft, Project administration, Writing—review and editing; AZ, Conceptualization, Data curation, Software, Formal analysis, Writing—review and editing; NTU, Conceptualization, Methodology, Writing—review and editing; AB, Data curation, This author's

contribution to this manuscript are acquisition of data and the management of the acquired data; MR, Software, Formal analysis; RJ, Resources, Methodology; MJM, Resources, Funding acquisition, Methodology; CLA, Resources, Supervision, Funding acquisition, Investigation, Writing—review and editing; TND, Conceptualization, Resources, Formal analysis, Supervision, Funding acquisition, Validation, Investigation, Writing—original draft, Project administration, Writing—review and editing

Author ORCIDs
Jae ook Kim, http://orcid.org/0000-0003-3500-2746
Trisha N Davis, http://orcid.org/0000-0003-4797-3152

## Additional files

### Supplementary files
• Supplementary file 1. Tables of pairwise statistical comparisons.

• Supplementary file 2. Table of yeast strains and plasmids used in this study.

• Source code 1. LabVIEW TIRF analysis step 1.

• Source code 2. LabVIEW TIRF analysis step 2.

• Source code 3. LabVIEW TIRF analysis step 3.

• Source code 4. Igor Pro compile TIRF data.

• Source code 5 Igor Pro analyze TIRF data.

• Source code 6. Igor Pro mean squared displacement TIRF analysis.

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
