## [Decision Letter]

Thank you for submitting your article "The Ndc80 complex bridges two Dam1 complex rings" for consideration by *eLife*. Your article has been favorably evaluated by Tony Hunter (Senior Editor) and three reviewers, one of whom, Andrea Musacchio, is a member of our Board of Reviewing Editors. The following individuals involved in review of your submission have agreed to reveal their identity: Stefan Westermann, and Kevin D Corbett.

The reviewers have discussed the reviews with one another and the Reviewing Editor has drafted this decision to help you prepare a revised submission.

Summary:

In this manuscript, the authors revisited the interaction of the Ndc80 complex, considered the main microtubule receptor at kinetochores, with the Dam1 complex, which is required for bi-orientation in *S. cerevisiae*. This interaction had been studied in quite some detail before, but the authors add an important new twist to the story by proposing that Ndc80 binds two Dam1 rings, rather than one as previously believed. The manuscript reports a nice mechanistic investigation that sheds new light on an important piece of molecular machinery. The *S. cerevisiae* kinetochore remains central for the advancement of kinetochore studies, and the dissection of its mechanism of action is of general interest for the field. While the reviewers are supportive, they also identified some contentious points that will need to be addressed in a revised version of the manuscript.

Essential revisions:

1) The definition of the A, B, and C region seems relatively clear for what concerns the Dam complex (defined by span of Ndc80 cross-linking pattern and containing most of the P sites previously identified), much less so for what concerns Ndc80. The original mutational analysis made use of 15-residue insertions, a serious perturbation that may be destabilizing. Are there conserved motifs that identify these Ndc80 regions and that could be mutated (substitutions, not deletion), singly and in combination, to reinforce the authors' contention that there are two separate binding sites? If this is not possible, are there at least a couple of lethal Ndc80 insertion mutants that fall outside of the predicted A, B, and C sites, and whose microtubule residence time could be measured? No change in residency time would be expected in this case if these mutations did not perturb the interaction with Dam1. Also, were mutations A, B, and C combined to see if they had an additive effect on residence time?

2) Subsection “Three Ndc80p regions interact with Dam1p, Ask1p, and Spc34p of the Dam1 complex”, first paragraph: more detail in the description of the cross-linking experiment should be provided: First, the authors should make clear that complete 4- and 10-protein Ndc80 and Dam1 complexes were used, and that the experiment was done in the presence of stabilized microtubules. Second, there must have been cross-links between Dam1 complex subunits and Nuf2 (which runs right alongside Ndc80); do these confirm or fail to confirm the model? All detected cross-links, also those with Nuf2 and possibly Spc24 and Spc25 should be reported.

3) The manuscript implies that two of the Ndc80 sites bind a single Dam1 ring, while a third Ndc80 site binds the second ring, but the reviewers where not clear whether it is sites A and B or B and C that function together. The authors should explicitly clarify their views on this point.

4) The reviewers believe that the in vivo experiments should be extended, at least by visualizing Dam1 in presence of single and double Ndc80 mutants that are kept alive by the Ndc80 degron allele, and where the relocalization of Dam1 from kinetochores to the spindle should be visible. This would be consistent with the work in vitro and give more credit to the hypothesis.

5) The EM experiments are interesting but have limitations. Ndc80 is not visible as being associated with the differently spaced rings. Doublets either with normal spacing or with increased spacing in the presence of Ndc80 seem to make up only a small fraction of the total number of Dam1 rings present in the sample. If the authors' claim is correct, Ndc80 should force the Dam1 rings into doublets when Ndc80 concentration is raised. This has not been tested. The authors should vary the stoichiometry of Ndc80 in the EM experiments to see if they can force complex assembly and a tighter distribution.

---

## [Author Response]

Essential revisions:

1) The definition of the A, B, and C region seems relatively clear for what concerns the Dam complex (defined by span of Ndc80 cross-linking pattern and containing most of the P sites previously identified), much less so for what concerns Ndc80. The original mutational analysis made use of 15-residue insertions, a serious perturbation that may be destabilizing. Are there conserved motifs that identify these Ndc80 regions and that could be mutated (substitutions, not deletion), singly and in combination, to reinforce the authors' contention that there are two separate binding sites? If this is not possible, are there at least a couple of lethal Ndc80 insertion mutants that fall outside of the predicted A, B, and C sites, and whose microtubule residence time could be measured? No change in residency time would be expected in this case if these mutations did not perturb the interaction with Dam1. Also, were mutations A, B, and C combined to see if they had an additive effect on residence time?

We understand your concern for how the Ndc80 five amino acid insertional mutants may be destabilizing and suggestion of circumventing this issue with substitution mutations. The insertional mutations in Ndc80p regions A, B, and C were previously characterized (Tien et al., 2013) and shown to form the Ndc80 complex in vivoand in vitro and bind microtubules in vitro similarly to the wild type Ndc80 complex. As defined in Table 1, each Ndc80p interaction region is quite long, spanning between 60 and 98 amino acids. These are largely coiled coil regions and no motifs are obvious beyond the coiled coil heptad repeat. Finding specific substitution mutations that disrupt the interaction between the Ndc80 and Dam1 complexes would be quite difficult.

We appreciate your alternative suggestion to test lethal Ndc80p insertional mutants that are outside of the interaction regions for binding microtubules and Dam1 complex. As shown in Figure 1—figure supplement 3, we tested two additional lethal Ndc80p insertional mutants at amino acid positions 219 and 652. Both of these Ndc80 complex mutants had microtubule and Dam1 binding characteristics similar to those of the wild type Ndc80 complex, demonstrating that the disrupted interaction between the Dam1 complex and Ndc80 complex with a mutation in A^ndc80p^, B^ndc80p^, or C^ndc80p^ is not an artifact of the five amino acid insertion. This data is presented in Figure 1—figure supplement 3 and subsection “Three Ndc80p regions interact with Dam1p, Ask1p, and Spc34p of the Dam1 complex”, last paragraph.

Thank you for your suggestion of combining mutations A, B, and C together. Instead of pursuing this in vitro experiment, we focused our efforts on making the combination mutants to test in vivo. Please see below.

2) Subsection “Three Ndc80p regions interact with Dam1p, Ask1p, and Spc34p of the Dam1 complex”, first paragraph: more detail in the description of the cross-linking experiment should be provided: First, the authors should make clear that complete 4- and 10-protein Ndc80 and Dam1 complexes were used, and that the experiment was done in the presence of stabilized microtubules. Second, there must have been cross-links between Dam1 complex subunits and Nuf2 (which runs right alongside Ndc80); do these confirm or fail to confirm the model? All detected cross-links, also those with Nuf2 and possibly Spc24 and Spc25 should be reported.

The requested additional details about the conditions of the cross-linking experiments have been added to the manuscript in the first paragraph of the subsection “Three Ndc80p regions interact with Dam1p, Ask1p, and Spc34p of the Dam1 complex”.

Thank you for asking about the cross-links between the Dam1 complex subunits (Dam1p, Ask1p, Spc34p) and Nuf2p. As predicted the Dam1 complex proteins form cross-links to Nuf2p. The Dam1p-Nuf2p cross-links correspond to the exact regions of Nuf2p predicted from the Dam1-Ndc80 cross-links, given the register of the coiled coil between Ndc80p and Nuf2p. These data are presented in Figure 1—figure supplement 2 and discussed in the last paragraph of the aforementioned subsection.

We agree with *eLife* and the reviewers on full presentation and transparency of our data. We have recently developed an online Protein Cross-Linking Database (ProXL) for easy visualization and sharing of cross-linking data (Riffle et al., 2016). This is an effective method for sharing large and complicated cross-linking/mass spectrometry data as it allows users full access to both the raw and processed data as well as providing comprehensive tools for analyzing the data. As stated in the Materials and methods section, the full cross-linking data referred to in the manuscript are available to the public: (http://proxl.yeastrc.org/proxl/viewProject.do?project_id=24). Please access the full cross-linking data by going to this link.

3) The manuscript implies that two of the Ndc80 sites bind a single Dam1 ring, while a third Ndc80 site binds the second ring, but the reviewers where not clear whether it is sites A and B or B and C that function together. The authors should explicitly clarify their views on this point.

We have included a paragraph (Discussion, fourth paragraph) and a figure (Figure 7) to clarify our views on this matter.

*4) The reviewers believe that the* in vivo *experiments should be extended, at least by visualizing Dam1 in presence of single and double Ndc80 mutants that are kept alive by the Ndc80 degron allele, and where the relocalization of Dam1 from kinetochores to the spindle should be visible. This would be consistent with the work* in vitro *and give more credit to the hypothesis.*

We appreciate the reviewers’ suggestion of this experiment. We think it is a very good hypothesis, given our results thus far. We have visualized the kinetochore and Dam1 complex with Mtw1-YFP and Dad4-CFP, respectively. Our new results suggest that the Dam1 complex becomes delocalized from kinetochores in the presence of Ndc80 with one or multiple mutations in regions A, B, and C. These results are presented in a new Figure 6 and in the second paragraph of the subsection “The Ndc80 mutants have defects in recruiting the Dam1 complex to the kinetochore”.

5) The EM experiments are interesting but have limitations. Ndc80 is not visible as being associated with the differently spaced rings. Doublets either with normal spacing or with increased spacing in the presence of Ndc80 seem to make up only a small fraction of the total number of Dam1 rings present in the sample. If the authors' claim is correct, Ndc80 should force the Dam1 rings into doublets when Ndc80 concentration is raised. This has not been tested. The authors should vary the stoichiometry of Ndc80 in the EM experiments to see if they can force complex assembly and a tighter distribution.

The EM experiments are set up such that the Dam1 complex rings do not fully decorate the microtubules. Nevertheless, 49% of the Dam1 ring pairs were found in closely spaced doublets in the presence of the wild-type Ndc80 complex. This demonstrates that half the Dam1 complex rings are bridged by wild-type Ndc80 complex. We agree that this is a very important point and have added it to the manuscript (subsection “The Ndc80 complex binds two Dam1 complex rings”, second paragraph).